# Evaluation of Antioxidant-Rich Mexican Oregano (*Lippia graveolens*) Infusion and Carvacrol: Impact on Metabolic Activity and Cytotoxicity in Breast Cancer Cell Lines

**DOI:** 10.3390/nu17193089

**Published:** 2025-09-28

**Authors:** Brian Enrique Rojo-Ruvalcaba, Montserrat Maldonado-González, Gabriela María Cálix-Rodríguez, Elia Herminia Valdés-Miramontes, Juan Florencio Gómez-Leyva, Teresa Arcelia García-Cobián, Pedro Ernesto Sánchez-Hernández, Andrea Carolina Machado-Sulbaran, Rocío Ivette López-Roa, Iván Balderas-León, Trinidad García-Iglesias

**Affiliations:** 1Programa de Doctorado en Ciencias en Biología Molecular en Medicina, Departamento de Biología Molecular y Genómica, Centro Universitario de Ciencias de la Salud (CUCS), Universidad de Guadalajara, Guadalajara 44340, Jalisco, Mexico; brian.rojo0999@alumnos.udg.mx; 2Centro de Investigación en Enfermedades Infectocontagiosas (CIEIC), Departamento de Microbiología y Patología, Centro Universitario de Ciencias de la Salud (CUCS), Universidad de Guadalajara, Guadalajara 44340, Jalisco, Mexico; montserrat.maldonado@academicos.udg.mx; 3Instituto de Cáncer en la Infancia y la Adolescencia (INICIA), Centro Universitario de Ciencias de la Salud (CUCS), Universidad de Guadalajara, Guadalajara 44340, Jalisco, Mexico; gabriela.calix2285@alumnos.udg.mx (G.M.C.-R.); pedro.shernandez@academicos.udg.mx (P.E.S.-H.); andrea.machado5223@academicos.udg.mx (A.C.M.-S.); 4Instituto de Investigaciones en Comportamiento Alimentario y Nutrición (IICAN), Centro Universitario del Sur (CUSUR), Universidad de Guadalajara, Ciudad Guzmán 49000, Jalisco, Mexico; eliav@cusur.udg.mx; 5Tecnológico Nacional de México-Instituto Tecnológico de Tlajomulco, DEPI, Tlajomulco de Zúñiga 45640, Jalisco, Mexico; jfgleyva@hotmail.com; 6Instituto de Terapéutica Experimental y Clínica (INTEC), Departamento de Fisiología, Centro Universitario de Ciencias de la Salud (CUCS), Universidad de Guadalajara, Guadalajara 44340, Jalisco, Mexico; arcelia.garcia@academicos.udg.mx; 7Laboratorio de Investigación y Desarrollo Farmacéutico (LIDF), Centro Universitario de Ciencias Exactas e Ingenierías (CUCEI), Universidad de Guadalajara, Guadalajara 44430, Jalisco, Mexico; rocio.lopez@academicos.udg.mx (R.I.L.-R.); ivan.balderas@academicos.udg.mx (I.B.-L.)

**Keywords:** Mexican oregano, anticancer agents, alternative treatments, breast cancer triple negative, antioxidant capacity, plant extracts, isolated compounds

## Abstract

**Background/Objectives:** The search for natural alternatives in breast cancer (BC) management has spurred interest in plant-derived extracts, particularly oregano variants and their bioactive compound carvacrol (Cv). However, Mexican oregano (*Lippia graveolens*) infusion (MoI) remains unexplored. This study aimed to chemically characterize MoI and compare its anticancer effects with Cv across BC cell lines, including aggressive triple-negative (TN) subtypes. **Methods:** MoI was analyzed for composition, antioxidant capacity (ABTS, DPPH, FRAP, total phenols/flavonoids), and phytochemical profile (FTIR, HPLC). Anticancer activity was assessed via MTT and LDH assays. **Results:** MoI exhibits strong antioxidant capacity and concentration-dependent antiproliferative effects, with IC50 values ranging from 0.08 to 0.18 mg/mL across BC lines, significantly higher (i.e., less cytotoxic) than Cv IC50 of 121–211 µM. Importantly, MoI displayed markedly lower cytotoxicity toward non-cancerous cells (IC50 0.18 mg/mL) compared to Cv (IC50 110 µM). **Conclusions:** While both agents reduced metabolic activity, Cv induced a more acute suppression. These findings position MoI as a promising, selective candidate for BC therapy, particularly for poor-prognosis subtypes like TN BC, warranting further mechanistic investigation.

## 1. Introduction

Cancer is a broad term encompassing a group of diseases characterized by uncontrolled growth and proliferation of abnormal cells. BC originates in mammary tissues and primarily affects lobules, ducts, and connective tissue; most cases arise in the ducts or lobules. According to the Global Cancer Observatory, BC accounted for 2,296,840 cases worldwide in 2022, representing 11.5% of all cancer diagnoses. It was also the leading cause of cancer-related mortality among women, with 666,103 deaths (15.4%). On the other hand, in Mexico, 31,043 cases of CaMa and 8195 deaths due to this disease were reported. These data confirm that CaMa is the type of cancer with the highest incidence and mortality in women, both in Mexico and worldwide [1,2,3].

In 2000, Perou et al. proposed a molecular classification of BC based on the expression of three key receptors critical for conventional treatment efficacy: estrogen receptor (ER), progesterone receptor (PR), and human epidermal growth factor receptor 2 (HER2). This classification defines four molecular subtypes with distinct therapeutic responses: luminal A (ER+ and PR+/−), luminal B (ER+, PR+/−, and HER2+), HER2-enriched (HER2+), and TN (ER−, PR−, and HER2−) [4]. Among these, TN is clinically recognized as the most aggressive subtype due to its poor response to conventional therapies, leading to lower overall survival and disease-free survival rates [4,5].

Natural products have played a fundamental role in traditional medicine throughout history. Since the World Health Organization endorsed the use of complementary therapies in 2004, interest in herbal medicine and the development of phytochemical-based anticancer agents has grown significantly. Bioactive compounds derived from plants, especially phenolic compounds such as polyphenols and flavonoids, have gained recognition not only for its potential for the development of anticancer treatments, but also for being a source of bioactive compounds with a safe therapeutic profile that are accessible and affordable [6,7,8,9]. *Lippia graveolens*, commonly known as Mexican oregano, is used as a condiment in many traditional Mexican dishes as well as raw material to produce cosmetics, drugs, and liquors. Oregano variants have been reported to have antioxidant potential and anti-inflammatory activity; these properties are attributed to their phytochemical profile, mainly to phenolic compounds present in both fat-soluble and water-soluble extracts [10,11].

However, limited evidence exists regarding the composition, antioxidant capacity, and anticancer potential of aqueous extracts from oregano variants, which are complex mixtures of potentially bioactive compounds, including Cv, leaving a gap in knowledge about the anticancer potential of MoI. Cv is a monoterpenoid phenol synthesized via the mevalonate pathway from acetyl-coenzyme A, which has attracted attention for its anticancer properties. It is one of the predominant compounds in oregano extracts. Despite being one of the most studied constituents of oregano extracts, there is still a lack of evidence regarding its molecular mechanisms in different types of cancer, such as CaMa [12,13,14].

The aims of this study was, first, to characterize the *Lippia graveolens* plant, as well as to evaluate the antioxidant capacity of the MoI, prepared with this endemic plant sample, with different tests (ABTS, DPPH, total phenols, total flavonoids, and FRAP), and second, to compare the anticancer effects of MoI and Cv at the level of cell metabolism and cytotoxicity against BC cell lines, luminal A (MCF-7), HER2-enriched (HCC-1954), and TN (MDA-MB-231). In addition, these tests were performed on HaCaT cells (immortalized keratinocytes that do not express oncogenes) to evaluate the effects of both treatments on non-cancerous cells.

## 2. Materials and Methods

### 2.1. Chemical and Reagents

The following reagents were used for this study: Cv ≥ 98% purity (499-75-2, Sigma Aldrich, St. Louis, MO, USA),dimethyl sulfoxide (DMSO, D8418, Sigma Aldrich, St. Louis, MO, USA), Dulbecco’s Modified Eagle Medium (DMEM, 11966025, Thermo Fisher Scientific^TM^, Waltham, MA, USA), Roswell Park Memorial Institute Medium (RPMI-1640, 11875119, Thermo Fisher Scientific^TM^, Waltham, MA, USA), phosphate-buffered saline (PBS, 10010023, Thermo Fisher Scientific^TM^, Waltham, MA, USA), fetal bovine serum (FBS, 16140071, Thermo Fisher Scientific^TM^, Waltham, MA, USA), penicillin/streptomycin (15140122, Thermo Fisher Scientific^TM^, Waltham, MA, USA), MTT (6494, Thermo Fisher Scientific^TM^, Waltham, MA, USA), and trypsin (25300054, Thermo Fisher Scientific^TM^, Waltham, MA, USA)All solvents and chemicals were of analytical grade.

### 2.2. Cell Lines and Culture Conditions

Cell lines were purchased from the American Type Culture Collection (ATCC; Manassas, VA, USA). Human keratinocyte-transformed and immortalized HaCaT cells and TN BC MDA-MB-231 cells were maintained in RPMI-1640, while luminal A MCF-7 and HER2 HCC-1954 BC cell lines were maintained in DMEM. Both mediums were supplemented with 10% FBS and 1% of penicillin/streptomycin. The cell cultures were kept in incubation under physiological conditions (37 °C, 95% humidity, and 5% CO_2_ saturation).

### 2.3. Plant Material

*Lippia graveolens* Kunth leaves were collected from the region of Totatiche, Jalisco, Mexico. This species was identified by Dr Pablo Carrillo-Reyes, and a voucher specimen (No. SIST-TRA-20250506) was deposited at the Instituto de Botánica de la Universidad de Guadalajara (IBUG) of Jalisco, Mexico (Appendix A).

### 2.4. Bromatological Analysis of L. graveolens

The chemical composition of the dried leaves of *L. graveolens* was determined by measuring moisture, ash, protein, fat, available carbohydrates, dietary fiber, and energy content, in duplicate, following the Normas Oficiales Mexicanas and the methods of the Association of Official Analytical Chemists (AOAC). Moisture content, calcination residue, and dietary fiber were determined by conventional gravimetric methods, using 5 g of sample for the first two determinations and 10 g for the dietary fiber analysis [15,16,17]. Crude protein content was determined by the modified Kjeldahl method, using 0.5 g of sample [18]. Crude fat content was evaluated from 1 to 2 g of sample by the Soxhlet extraction method [19]. Reducing carbohydrates were determined from 10 to 15 g of sample using the volumetric method with Fehling’s solution according to NOM-086-SSA1-1994 [17]. Energy content (kcal) was calculated according to the AOAC method of analysis for nutrition labeling, using general factors [20].

### 2.5. Aqueous Extraction of Mexican Oregano (Lippia graveolens Kunth)

An amount of 3 mL of injectable water was heated to boiling. After removing from heat, 69 mg of ground oregano leaves were added. The mixture was covered with aluminum foil to prevent loss of volatiles and left at room temperature for 10 min. It was then filtered through a 0.22 µm syringe filter and stored in a microtube for analysis.

### 2.6. Preparation of Cv Solution

Cv isolate was mixed with DMSO (<0.05% final concentration) and culture medium to obtain a 1000 µM solution.

### 2.7. Antioxidant Activity Protocols (DPPH and ABTS) in MoI

In brief, a total of 225 μL of ABTS (2,2′-azinobis-3-ethylbenzothiazoline-6-sulfonic acid) solution was placed in each microplate well, and an initial reading was taken at 734 nm, which served as the initial absorbance datum. Subsequently, 75 μL of 1:1000 diluted MoI (with deionized water) was added, and the microplate was incubated for 5 min with stirring. After which, the absorbance was measured at 734 nm. A Trolox (6-hydroxy-2,5,7,8-tetramethylchroman-2-carboxylic acid) standard curve was performed to express the results as mMEQ Trolox per gram of dried sample. Results are expressed as mean ± standard deviation of three independent experiments performed in triplicate.

For the DPPH (1,1diphenyl-2-picrylhydrazyl) method, 150 µL of the 1:1000 diluted MoI (with deionized water) was placed in a 96-well plate, and 150 µL of 1010 µM methanolic solution of DPPH was added and allowed to react in the dark at room temperature for 30 min. A Trolox standard curve was performed to express the results as mMEQ Trolox per gram of dried sample. The absorbance was measured at 517 nm (BioTek-SynergyHTX, Winooski, VE, USA). Results are expressed as mean ± standard deviation of three independent experiments performed in triplicate.

### 2.8. Determination of Total Phenolic Content in MoI

Total phenolic content of MoI was determined by the Folin–Ciocalteu method. An amount of 200 μL of distilled water was mixed with 250 μL of Folin–Ciocalteu solution (1 N) and 50 μL of MoI (1:10) or different concentrations (20–180 g/mL) of gallic acid (used as a standard). After incubation for 5 min at room temperature, 500 µL of sodium carbonate solution (15% Na_2_CO_3_) was added, the solution was mixed thoroughly, and incubated for 15 min in the dark at 45 °C. Then, 200 μL of each reaction was measured at 760 nm (BioTek-SynergyHTX, Winooski, VE, USA). All results were expressed as milligram of gallic acid equivalent (GAE) per gram of dried sample (mgGAE/g). Results are expressed as mean ± standard deviation of three independent experiments performed in triplicate.

### 2.9. Determination of Total Flavonoids in MoI

Briefly, 100 µL of MoI was mixed with 200 µL of 1 M potassium acetate, 200 µL of aluminum chloride solution (10% AlCl_3_), and 500 µL of 80% ethanol. The mixture was shaken vigorously and incubated for 40 min at room temperature. The absorbance was measured at a wavelength of 415 nm (BioTek-SynergyHTX, Winooski, VE, USA). A calibration curve was generated with quercetin as the standard. Results were expressed as mg Eq quercetin per gram of dried sample. Results are expressed as mean ± standard deviation of three independent experiments performed in triplicate.

### 2.10. FRAP in MoI

An amount of 40 μL of MoI or standard was mixed with 250 μL of FRAP reagent (25 mL 300 mM sodium acetate, 1.6% acetic acid; 2.5 mL TPTZ (2, 4, 6-tri (2-pyridyl)-s-triazine), 40 mM hydrochloric acid). An amount of 240 μL of distilled water was used as a blank and FeSO_4_ was used as a standard for the calibration curve. The absorbance was determined at 593 nm (BioTek-SynergyHTX, Winooski, VE, USA). Results were expressed as millimoles of Fe^2+^. Results are expressed as mean ± standard deviation of three independent experiments performed in triplicate.

### 2.11. FTIR Measurement

The MoI was analyzed using FTIR spectroscopy within the 4000–400 cm^−1^ range, utilizing a PerkinElmer 400 FTIR spectrometer (PerkinElmer, Waltham, MA, USA). A background spectrum was recorded prior to measurement to eliminate environmental interference. For analysis, MoI was placed on the sample holder of the ATR (Attenuated Total Reflectance) crystal, forming a thin film. The instrument performed 32 scans, which were averaged to generate the final spectrum. The results were displayed as transmittance vs. wavenumber (cm^−1^) for functional group identification.

### 2.12. HPLC Analysis of MoI

HPLC was performed to identify and quantify the major compounds in MoI including Cv, thymol, pinocembrin, and galangin, using two chromatographic methods. Cv and thymol were analyzed with an Agilent 1120 Compact HPLC system (Agilent, CA, USA) and UV detection at 274 nm. Separation was performed on a Phenomenex Luna C18 column (Phenomenex, Torrance, CA, USA) (250 × 4.6 mm, 5 µm) using a mobile phase of water with 0.05% trifluoroacetic acid (A) and acetonitrile (B). The gradient was 90:10 (A/B) at 0 min, 70:30 at 3 min, 30:70 at 20 min, and 70:30 at 25 min. The flow rate was 0.7 mL/min, oven temperature 40 °C, and injection volume 10 µL. Pinocembrin and galangin were analyzed with a Thermo Scientific Dionex ICS-50000 HPLC-DAD system (Thermo Scientific, Waltham, MA, USA), quantified using detection at 245, 270, 290, and 515 nm. A Gemini C18 column (250 × 4.6 mm, 5 µm) was used with a mobile phase of 0.5% formic acid in water (A) and acetonitrile (B). The gradient was: 2–12% B (0–15 min), 12% B (15–23 min), 12–40% B (23–46 min), 40–90% B (46–71 min), and 90% B (71–75 min), at 1.0 mL/min. Samples (15 µL) were filtered (0.22 µm) and diluted 1:1 in methanol–water (50:50) with 0.5% formic acid. Calibration curves were prepared using ≥98% purity standards of Cv, thymol, pinocembrin, and galangin. Data were processed with EZChrom Elite software version 3.3.2.

Thymol and Cv were quantified using external calibration curves with the equations y = 0.006x − 52.1066 (R^2^ = 0.9912) and y = 0.0064x − 0.0724 (R^2^ = 0.9943), respectively. Pinocembrin and galangin were also confirmed and quantified using standard curves with equations y = 0.2137x + 0.365 (R^2^ = 0.9915) and y = 2.8792x − 0.798 (R^2^ = 0.980). The strong correlation coefficients across all models support reliable quantification and confirm the presence of these bioactive flavonoids in the extract.

### 2.13. MoI Time Response Curve

Since there are no previous reports on the use of MoI on cancer cells, we evaluated the effect at the metabolic level of different doses of the infusion for 24, 48, and 72 h in MDMA-MB-231 cells to establish an exposure time for further testing, whereas for Cv treatment we referred to the background of Arunasree et al. in MDA-MB-231 cells [16]. Briefly, 1.5 × 10^4^ cells/well were seeded in 96-well plates. After 24 h of incubation, MoI was added in different concentrations, untreated groups received culture medium, and cells were incubated for the corresponding time (24, 48, or 72 h). Next, MTT (5 mg/mL in PBS) was added, and cells were incubated for 4 h to allow formazan production. These crystals were dissolved using a solution prepared with sodium dodecyl sulfate (28312, Thermo Fisher Scientific^TM^, Waltham, MA, USA) and dimethylformamide (227056, Sigma Aldrich, St. Louis, MO, USA) at pH 4.6, followed by a final incubation of 18–24 h. Absorbance was measured at 570 nm (BioTek-800TS, Winooski, VE, USA), with untreated cells considered as 100% formazan production. Results are expressed as mean ± standard deviation of three independent experiments performed in triplicate.

### 2.14. Evaluation of Cell Metabolism (MTT Assay)

Cells were seeded in 96-well plates at a density of 1.5 × 10^4^ cells/well. After 24 h of incubation, increasing concentrations of MoI or Cv solution were added, while untreated groups received culture medium. Cells were incubated 24 h with Cv and 48 h with MoI. Subsequently, MTT (5 mg/mL in PBS) was added, and cells were incubated for 4 h. Formazan crystals were dissolved using a solution prepared with sodium dodecyl sulfate (28312 Thermo Fisher Scientific^TM^, Waltham, MA, USA) and dimethylformamide (227056 Sigma Aldrich, St. Louis, MO, USA) at pH 4.6, followed by a final incubation of 18–24 h. Absorbance was measured at 570 nm (BioTek-800TS, Winooski, VE, USA), with untreated cells considered as 100% formazan production. A dose–response curve was generated for each cell line under both treatments independently, and half maximal inhibitory concentration (IC50) values were determined using the four-parameter logistic model [21]. The IC50 values obtained were subsequently used for the cytotoxicity assay. Results are expressed as mean ± standard deviation of three independent experiments performed in triplicate.

### 2.15. Cytotoxicity Assay (LDH in Supernatant)

Cytotoxicity was assessed using the CyQUANT™ LDH Cytotoxicity Assay Kit (C20300, Thermo Fisher Scientific^TM^, Waltham, MA, USA), following the manufacturer’s instructions. Briefly, 1.5 × 10^4^ cells were seeded in a 96-well plate and treated with MoI, Cv, or ultrapure water (for the spontaneous LDH activity control) in a final volume of 100 µL. Cells were incubated for 24 h in the Cv group and 48 h in the MoI group. Subsequently, 10 µL of 10× Lysis Buffer was added to the maximum LDH control group and incubated for 45 min. Then, 50 µL of supernatant from each well was transferred to a new 96-well plate, followed by the addition and mixing of 50 µL of reaction mixture. After 30 min of incubation at room temperature, 50 µL of stop solution was added and mixed. Finally, absorbance was measured at 490 nm with correction at 690 nm (BioTek-SynergyHTX, Winooski, VE, USA). Results are expressed as mean ± standard deviation of three independent experiments performed in triplicate. Cytotoxicity was calculated using the following formula:$$\%\;Cytotoxicity\;=\;\left(\frac{Compund\;treated\;LDH\;actitity\;-\;Spontaneous\;LDH\;activity}{Maximun\;LDH\;activity\;-\;Spontaneous\;LDH\;activity}\right)\times \;100$$

### 2.16. Statistical Analysis

Results were expressed as means ± SD, Student’s *t*-test (cytotoxicity assay), ANOVA, and Dunnett’s post hoc test (MoI time response curve and evaluation of cell metabolism) was performed using R version 4.4.3, and a *p* value of <0.05 was considered statistically significant.

The following Figure 1 summarizes the workflow followed for this project.

## 3. Results

### 3.1. Nutritional Composition of L. graveolens

Table 1 shows the results obtained from the bromatological analysis of the *L. graveolens* sample.

### 3.2. Antioxidant Capability of MoI

These results show that MoI has significant total antioxidant capacity, demonstrating that it is an extract rich in compounds with potential biological activity (Table 2).

### 3.3. MoI Chemical Composition by FTIR

Figure 2 shows the FTIR spectrum of MoI and allows us to confirm the presence of functional groups characteristic of molecules with recognized antioxidant capacity, such as phenols, polyphenols, and flavonoids.

### 3.4. High-Performance Liquid Chromatography (HPLC) Analysis of MoI

Figure 3 and Figure 4 show the HPLC determinations of Cv, thymol, galangin, and pinocembrin in MoI. These results confirm that MoI contains these compounds, which are being studied for their anticancer potential.

### 3.5. MoI Dose–Response Curve

Treatment with MoI for 24 h does not modify the metabolic activity of MDA-MB-231 cells. Instead, the infusion significantly decreases formazan production after 48 h (0.08 mg/mL, 83.09% ± 5.17%; 0.09 mg/mL, 88.20% ± 1.81%; and 0.1 mg/mL, 82.66% ± 5.68%) and this effect is similar at 72 h (0.08 mg/mL, 86.23% ± 3.48%; 0.09 mg/mL, 90.65% ± 1.31%; and 0.10 mg/mL, 89.88% ± 2.63%). These results show the attenuated anticancer potential of MoI against CaMa cell lines of the most aggressive subtype (Figure 5).

### 3.6. Cell Metabolism Assay

The differential treatment protocols employed—24 h Cv exposure versus 48 h complex aqueous extract (MoI) treatment—demonstrate the fundamental pharmacodynamic distinction between isolated monoterpenes and multicomponent botanical preparations. Carvacrol exhibits rapid, aggressive anticancer effects through mitochondria-mediated apoptosis, ROS generation, and disruption of multiple signaling pathways, including PI3K/AKT and MAPK, within abbreviated timeframes. In contrast, complex extracts rich in antioxidant molecules operate through attenuated, multitargeted mechanisms that require extended exposure periods to achieve therapeutic efficacy, reflecting their more balanced approach to cellular redox modulation and their synergistic component interactions. The clinical translation implications of these contrasting therapeutic profiles represent a critical consideration for anticancer drug development. While isolated Cv demonstrates superior in vitro potency, its aggressive cytotoxic nature presents significant challenges for safe systemic administration, including potential dose-limiting toxicities and complications in patient management protocols. Conversely, the attenuated activity profile of complex extracts, despite requiring prolonged treatment durations, offers enhanced clinical translatability through improved biocompatibility, reduced off-target effects, and better patient tolerance. This paradigm reinforces the therapeutic potential of natural product research focused on standardized botanical preparations that can harness synergistic phytochemical interactions while maintaining the safety margins necessary for successful clinical implementation in cancer management. Treatment of HaCaT cells with MoI significantly reduced formazan production starting at 0.10 mg/mL (84.51% ± 4.50%), with a dose-dependent effect up to 0.30 mg/mL (9.50% ± 3.82%). Similarly, Cv showed a dose-dependent response, beginning at 50 µM (54.49% ± 6.42%) and decreasing to 22.61% ± 4.15% at 300 µM.

In MCF-7 cell line, MoI treatment showed a significant reduction in formazan production from the 0.10 mg/mL dose onward (34.67% ± 6.31%), with the effect dropping to 13.70% ± 3.91% at 0.30 mg/mL. Cv also reduced cellular metabolism from the first dose (50 µM, 81.00% ± 5.86%), with the effect increasing as the dose increased, reaching 27.55% ± 2.77% at 500 µM.

HCC-1954 cells treated with MoI showed significant results starting from the second dose at 0.15 mg/mL (74.56% ± 3.59%), reaching a low of 47.60% ± 6.92% at 0.20 mg/mL. In contrast, Cv demonstrated a dose-dependent response from the first dose (50 µM, 85.18% ± 5.59%), decreasing to 40.95% ± 3.04% at 200 µM.

Finally, MDA-MB-231 cell line showed a significant dose-dependent response to both treatments. For MoI, the effect started at 0.10 mg/mL (74.71% ± 5.36%), decreasing to 41.56% ± 5.89% at 0.30 mg/mL. For Cv, the response began at 50 µM (85.76% ± 4.14%), with the lowest formazan production recorded at 500 µM (25.15% ± 3.25%). In general, Cv caused a drastic reduction in metabolic activity in the four cell lines, compared to MoI, which showed gradual results. These results are presented in Figure 6 and Figure 7.

### 3.7. IC50 Calculation

The IC50 values calculated for each cell line under each treatment are presented in Table 3.

Analyzing these results, we can observe contrasting behaviors: the more aggressive the molecular subtype of BC, the lower the dose of Cv required; conversely, the more aggressive the BC subtype, the higher the dose of MoI required to reach the IC50. Furthermore, MoI is less aggressive against non-cancerous cells, as evidenced by the fact that the control cell line requires the highest dose of MoI and the lowest dose of Cv for their respective IC50.

### 3.8. Cytotoxicity of MoI and Cv (IC50)

As shown in Figure 8, the IC50 Cv concentrations induced approximately 50% cytotoxicity, with values of 49.85% ± 6.53% in MDA-MB-231 cells, 51.26% ± 9.42% in HCC-1954 cells, 53.84% ± 8.59% in MCF-7 cells, and 57.66% ± 10.05% in HaCaT cells. In contrast, the IC50 concentrations of MoI resulted in a lower cytotoxicity of approximately 15%, with specific values of 11.68% ± 1.97% in MDA-MB-231 cells, 21.94% ± 3.25% in HCC-1954 cells, 15.43% ± 0.15% in MCF-7 cells, and 12.07% ± 2.08% in HaCaT cells.

In summary, MoI showed lower cytotoxicity compared to Cv in all cell lines, as well as being less aggressive than Cv against the control cell line.

## 4. Discussion

Both oregano extracts and Cv have been extensively studied due to their diverse biological activities, which postulate them as potential therapeutic agents for the management of different diseases, including cancer. Among their properties, their antioxidant activity stands out, as well as their antiproliferative and cytotoxic effects, characteristics that make them promising options for oncological treatment [22,23,24].

The bromatological analysis performed on the plant sample used in this study can be compared with the data reported by the U.S. Department of Agriculture (USDA) for dried oregano. Compared to this reference, our sample presented 12.3% less moisture, 15.3% less ash, 24% less protein, 2.3% less lipids, but showed increases of 51.9% in available carbohydrates and 70.1% in total sugars, along with a reduction of 9.1% in dietary fiber. These differences can be attributed to several factors, such as soil type and fertility, pH, growing conditions (temperature, light, humidity), irrigation regime, fertilization, pest control, harvest timing, and drying methods. Each of these factors significantly influences the chemical composition of the plant sample and, consequently, its biological activity [25].

Regarding the antioxidant capacity of *L. graveolens*, it has been demonstrated that both the plant matrix and its extracts are rich in antioxidant compounds, which confers a high capacity to neutralize free radicals [23,26]. The results obtained in the present study support this claim, since it was observed that MoI exhibited potent total antioxidant activity in ABTS and DPPH radical neutralization assays, as well as in the quantification of total phenols, total flavonoids, and FRAP. Similar results have been reported in the literature. Mahomoodally et al. (2019) evaluated the antioxidant capacity of aqueous and methanolic extracts of *Origanum onites* (Greek oregano), finding that the aqueous extract presented a higher antioxidant capacity compared to the methanolic one, which was attributed to a higher concentration of phenolic compounds [27]. In contrast, Simirgiotis et al. (2020) analyzed a lipid extract of *Origanum vulgare* (oregano species native to the Mediterranean), finding a lower amount of phenolic compounds and, consequently, a lower antioxidant capacity against ABTS and DPPH radicals [28]. Similarly, Kogiannou et al. (2013) compared the antioxidant capacity of six herbal infusions of plants originating from Greece, including a species of oregano, where they observed that this infusion had the best antioxidant capacity in all the techniques they used (total phenols, total flavonoids, DPPH, and FRAP), compared to other plants with recognized antioxidant capacity such as chalk marjoram, pink savory, mountain tea, pennyroyal, and chamomile [29]. These findings reinforce the evidence that extracts from oregano variants, especially in infusion form, have a high antioxidant potential. The antioxidant potential of MoI could be related to its anticancer activity, since there is a correlation between the amount and type of phenolic compounds and their cytotoxicity. It has been proposed that certain functional groups, such as carbonyls and free hydroxyls, facilitate the formation of ortho-diphenolic radicals (two hydroxyl groups attached to a benzene ring in adjacent positions), which could chelate transition metals involved in redox processes, thus affecting the viability of cancer cells [30,31]. This suggests the need to continue investigating their role in the management of chronic degenerative diseases, such as BC.

Regarding the compounds present in MoI, the FTIR spectrum analysis exhibited the presence of key phytochemical constituents known for their biological activity. The broad O–H stretching band is indicative of phenolic hydroxyl groups, typically found in flavonoids and phenolic acids [32]. Asymmetric and symmetric C–H stretching of aliphatic chains is associated with terpenoids like Cv and p-cymene [33,34]. Aromatic skeletal vibrations are observed, typically assigned to flavonoids, Cv, and thymol derivatives [32]. C–O stretching corresponds to phenolic ethers and alcohols, while the presence of C–O–C stretching confirms the presence of glycosidic and ester bonds, reinforcing the detection of flavonoid glycosides and phenolic esters [32,35]. Additional bands around ~947 and ~808 cm^−1^ are characteristic of out-of-plane aromatic C–H bending vibrations, confirming substituted aromatic structures typical of essential oil components [33]. These spectral features strongly correlate with the antioxidant and antiproliferative effects observed in the biological assays, confirming that MoI contains multiple bioactive constituents with therapeutic relevance.

HPLC profiling of MoI revealed thymol and Cv as the predominant non-polar constituents, eluting at 18.63 and 19.20 min, respectively. Quantitative analysis showed a markedly higher concentration of thymol (51.48 mg/mL) compared to Cv (0.0607 mg/mL). The method demonstrated adequate sensitivity, with limit of detection (LOD), and limit of quantification (LOQ) of 0.08/0.25 µg/mL for thymol and 0.07/0.22 µg/mL for carvacrol. A series of early-eluting peaks (0–10 min) likely correspond to hydrophilic phenolic acids and flavonoid glycosides, which contribute to the complex chemical matrix of the infusion and may enhance its antioxidant activity through synergistic interactions.

In the flavonoid fraction, galangin (126.0 µg/mL) was identified as the predominant compound, while pinocembrin was present at significantly lower levels (1.001 µg/mL). The method sensitivity also covered these flavonoids, with LOD, and LOQ of 0.05/0.15 µg/mL for galangin and 0.10/0.30 µg/mL for pinocembrin. These flavonoids contribute to the strong antioxidant capacity of MoI, as confirmed by ABTS, DPPH, and FRAP assays, and provide a chemical basis for its observed biological effects, including reduced metabolic viability in BC cell lines.

Additionally, prior studies by Anvarbatcha et al. and Vahitha et al. demonstrate that thymol, also present in the extract, exhibits selective cytotoxicity against MDA-MB-231 and MCF-7 BC cells. Thymol induces G_0_/G_1_ cell-cycle arrest, linked to downregulation of Cyclin D1. Both studies confirm thymol’s ability to trigger apoptosis: Anvarbatcha et al. reported increased p53 expression, suppression of Bcl-xL, and activation of caspases-9 and -3, along with elevated reactive oxygen species (ROS), proposed as the central mechanism driving both cell-cycle arrest and apoptosis. Vahitha et al. observed reduced PCNA expression, further supporting thymol’s pro-apoptotic effect [36,37].

Galangin induces apoptosis via the intrinsic mitochondrial pathway and inhibits the PI3K/Akt signaling cascade, a key driver of cancer cell survival, proliferation, and metastasis, with particular efficacy against TN BC cells [38,39,40]. Pinocembrin similarly targets BC cells (MCF-7 and MDA-MB-231), triggering G_2_/M-phase cell-cycle arrest and apoptosis through caspase-3/9 activation, BAX upregulation, and Bcl-2 downregulation. Like galangin, it suppresses the PI3K/Akt pathway, thereby reducing cancer cell migration and invasion. Importantly, pinocembrin shows selective toxicity toward malignant cells, with significantly lower effects on nonmalignant mammary epithelial cells [41].

Collectively, the presence of these anticancer bioactives in the infusion matrix underscores the concept that the diversity of phytochemicals within a complex extract may confer enhanced therapeutic benefits for BC management.

On the other hand, treatment with Cv differentially affected metabolic activity across cell lines: the TN BC cell line MDA-MB-231 was most sensitive (IC50 = 121 µM), followed by HER2+ HCC-1954 (IC50 = 123 µM), while luminal A MCF-7 was less responsive (IC50 = 211 µM). An important finding was that the non-cancerous HaCaT keratinocyte line showed the highest sensitivity (IC50 = 110 µM), underscoring the need to evaluate selectivity using healthy cell models.

These results align with Elshafie et al. (2017), who found that Cv was the most potent cytotoxic compound from *Origanum vulgare* against HepG2 liver cancer cells (IC50 = 48 mg/L ≈ 318 µM), significantly reducing their metabolic activity (12%) compared to non-cancerous HEK293 kidney cells (38% viability), this confirms Cv strong anticancer potential, even outperforming other oregano compounds [22]. Both prior studies and the current findings confirm that Cv significantly reduces viability in various cancer types, including hepatocellular carcinoma and aggressive BC subtypes. Further evidence comes from Elbe et al. (2020), who demonstrated that Cv, along with thymol, exerts dose-dependent cytotoxicity in ovarian cancer cells (SKOV-3), with IC50 values of 322.50 µM for Cv and 316.08 µM for thymol after 24 h. Although thymol showed slightly greater potency in this model, both compounds displayed strong anticancer activity [42]. These results contrast with those presented by Elshafie et al. in 2017, who found that Cv exhibited greater cytotoxic potential compared to other major *Origanum vulgare* compounds, including thymol [22]. The differences may be attributed to the different cell lines used in each study. Despite this, Cv demonstrated significantly high cytotoxic potential, supported by the results presented in the present study, which evidences its efficacy in reducing the metabolic activity of different cancer cell lines, including BC, and consequently support the importance of continuing its study to assess the possibility of its use in the management of neoplastic diseases.

Concerning MoI, it was found that as the aggressiveness of the molecular subtype of the cell line increases, so does the dose required to reach IC-50. Furthermore, the control cell line (HaCaT) required the same infusion dose as the triple-negative BC cells, suggesting that MoI has a less aggressive effect on non-cancerous cells compared to the isolated Cv compound. These findings are consistent with those reported by Tuncer et al. in 2013 [43]. In their study, they treated three BC cell lines (MCF-7, MDA-MB-468, and MDA-MB-231), using an aqueous extract of *Origanum acutidens* (Turkish oregano) and assessed the impact on metabolic activity using MTT assays. As in our research, treatment efficacy waspresented in a dose-dependent manner. Together, these reports and our results support that oregano extracts represents a good alternative treatment against BC cell lines, including the most aggressive subtypes [43]. Furthermore, in 2018, Makrane et al. investigated the effect of infusion of a Moroccan species of oregano in HT-29 colon cancer cells and MDA-MB-231 BC cells and evaluated its impact on metabolic activity by WST1 assay, finding that the treatment reduced metabolic activity in a dose-dependent manner in both cell lines. They observed that triple-negative BC cells were more susceptible to treatment, with an IC50 ranging from 30.90 to 87.09 µg/mL, while in HT-29 cells this range ranged from 50.11 to 158.48 µg/mL [44]. These results are relevant to our study, as we also employed an infusion as a treatment in a TN BC cell line, providing a close reference to contrast our results. Based on this study and our findings, we can conclude that oregano extracts, including MoI, significantly reduce the metabolic activity of BC cell lines, even in the TN subtype, and that this effect occurs in a dose-dependent manner.

The results obtained in this study indicate that Cv exhibits greater cytotoxicity compared to MoI (~50% and ~15%, respectively), which suggests that while infusion may reduce cellular metabolism, the isolated Cv compound exhibits greater cytotoxicity. These results may be due to the fact that MoI is a complex mixture of phytochemicals (for example, Cv, tymol, galangine, and pinocembrine), where Cv represents a fraction of the extract and has already been reported as a potent anticarcinogen on its own. Despite the limited availability of studies on oregano cytotoxicity, the existing evidence concurs that its extracts, regardless of extraction method, possess activity against BC cells, including the most aggressive subtypes. This could be due to antioxidant-rich treatments profoundly modulating BC cell signaling pathways through complex mechanisms that both exploit and counteract oxidative stress dynamics. For example, polyphenols primarily target key oncogenic pathways such as NF-κB, PI3K/AKT/mTOR, and MAPK cascades. The therapeutic effects operate through a dual mechanism: at moderate concentrations, antioxidants selectively induce apoptosis in cancer cells by disrupting survival signaling while simultaneously blocking ROS-mediated proliferation pathways. Specifically, well-studied compounds like curcumin inhibit STAT3 phosphorylation and NF-κB transcriptional activity, leading to reduced expression of anti-apoptotic proteins and metastatic markers like MMP-9 and VEGF. Additionally, these treatments enhance chemosensitization by activating Nrf2-mediated antioxidant response elements, which paradoxically increases therapeutic efficacy while protecting normal cells from oxidative damage. The PI3K/AKT pathway, frequently hyperactivated in BC through ROS-mediated PTEN inactivation, becomes particularly vulnerable to polyphenol intervention, as these compounds restore tumor suppressor function and promote apoptotic cell death. This multifaceted approach allows antioxidant treatments to simultaneously target cancer cell survival mechanisms, reduce metastatic potential, and overcome drug resistance while maintaining selectivity for malignant cells over healthy tissue [45,46,47,48,49].

Regarding the limitations of this study, it should be acknowledged that an in vitro study represents a first phase of research, and our design does not allow us to establish causality between the antioxidant-rich MoI or Cv and their effects on cancer cell metabolism. Therefore, our results lay the foundation for future research to continue advancing to stages such as experimentation in models that allow for the evaluation of metabolic and molecular mechanisms of MoI in organisms, in order to continue assessing the feasibility of its use for the management of BC patients.

## 5. Conclusions

This study offers new insights into MoI composition, antioxidant properties, and its possible in vitro effects on metabolic activity and cytotoxicity across different molecular subtypes of BC, laying the groundwork for future research into its molecular mechanisms, and interactions with conventional therapy, which are fundamental aspects for MoI clinical validation. Given the poor prognosis of triple-negative BC, natural extracts like MoI represent a promising avenue for development of novel cancer therapies. Although these results are encouraging, further studies are needed to ensure the feasibility of its use in patients.

## Figures and Tables

**Figure 1 nutrients-17-03089-f001:**
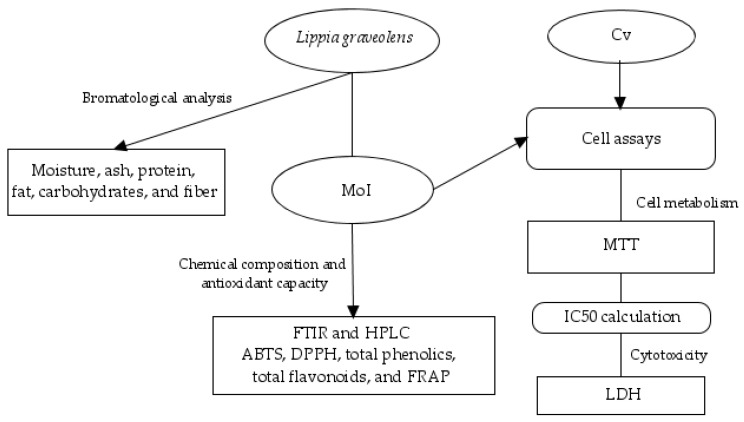
Summary of the experimental design.

**Figure 2 nutrients-17-03089-f002:**
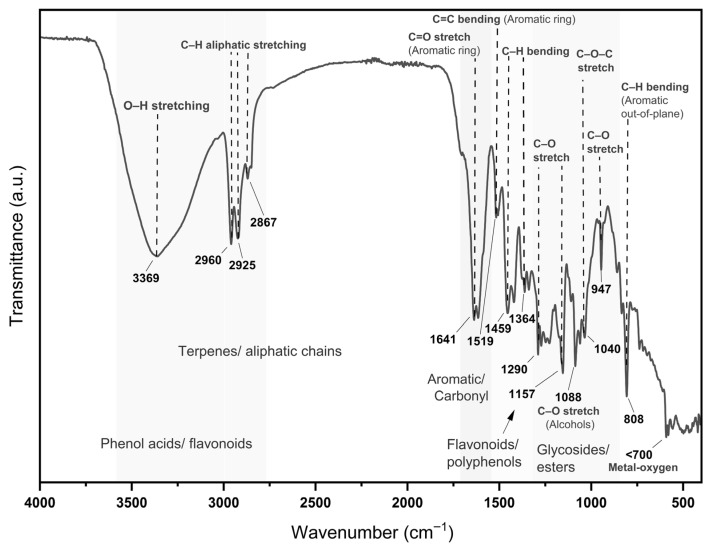
FTIR spectrum of Lippia graveolens infusion showing bands for phenols (~3369 cm^−1^), terpenes (~2925–2960 cm^−1^), aromatics (~1641 cm^−1^), and glycosides (~1040 cm^−1^), indicating the presence of bioactive compounds.

**Figure 3 nutrients-17-03089-f003:**
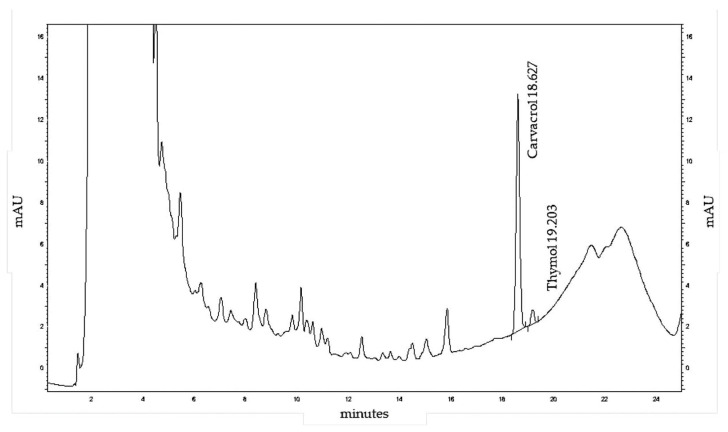
HPLC chromatogram of Mexican oregano (*Lippia graveolens* Kunth) extract at 274 nm, showing major peaks for Cv (18.63 min) and thymol (19.20 min), along with various minor phenolic compounds.

**Figure 4 nutrients-17-03089-f004:**
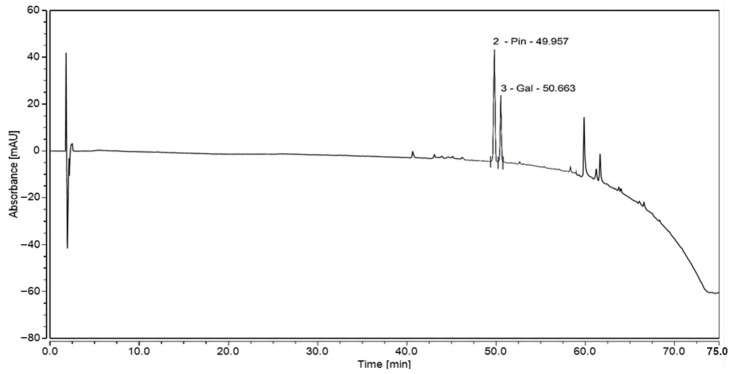
HPLC chromatogram of aqueous oregano leaf extract recorded at 290 nm. Pinocembrin (49.96 min) and galangin (50.68 min) were identified based on retention time and UV spectra.

**Figure 5 nutrients-17-03089-f005:**
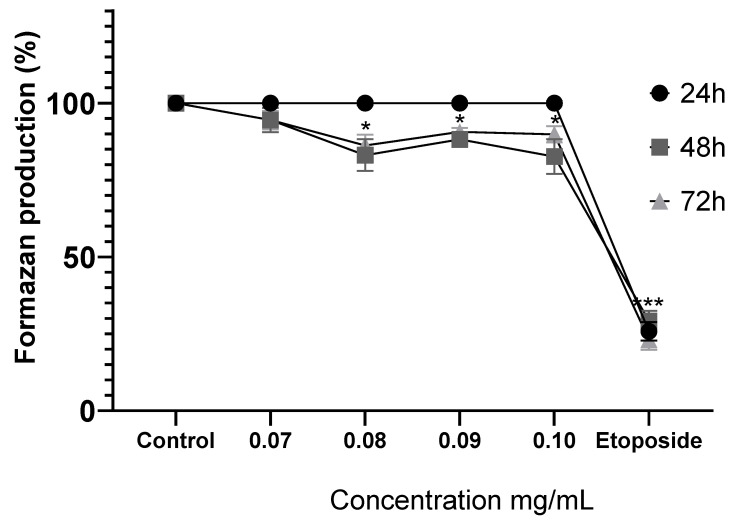
Comparison of formazan production in MDA-MB-231 cells after exposure to MoI for 24, 48, and 72 h. Etoposide (0.18 mg/mL = 305.8 µM) was used as a cell death control. Determination was performed by MTT assay, and percentages were calculated in comparison to the untreated group, which was set as 100% formazan production. Results are expressed as mean ± standard deviation from three independent experiments performed in triplicate. Statistical analysis between groups was performed using ANOVA and Dunnett’s post hoc test (* = *p* < 0.05; *** = *p* < 0.001, compared with control).

**Figure 6 nutrients-17-03089-f006:**
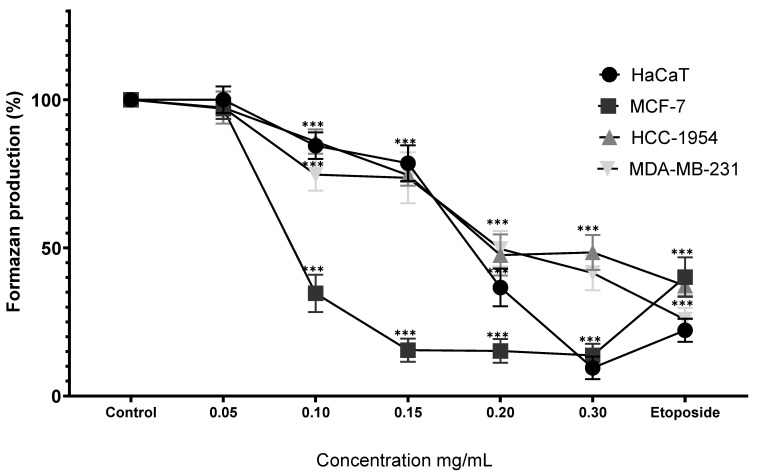
Percentage of formazan produced after treatment with MoI for 48 h. Etoposide (0.18 mg/mL = 305.8 µM) was used as cell death control. Determination was performed by MTT assay, and percentages were calculated in comparison to the untreated group, which was set at 100% formazan production. Results are expressed as mean ± standard deviation from three independent experiments performed in triplicate. Statistical analysis between groups was performed using ANOVA and Dunnett’s post hoc test (*** = *p* < 0.001, compared with control).

**Figure 7 nutrients-17-03089-f007:**
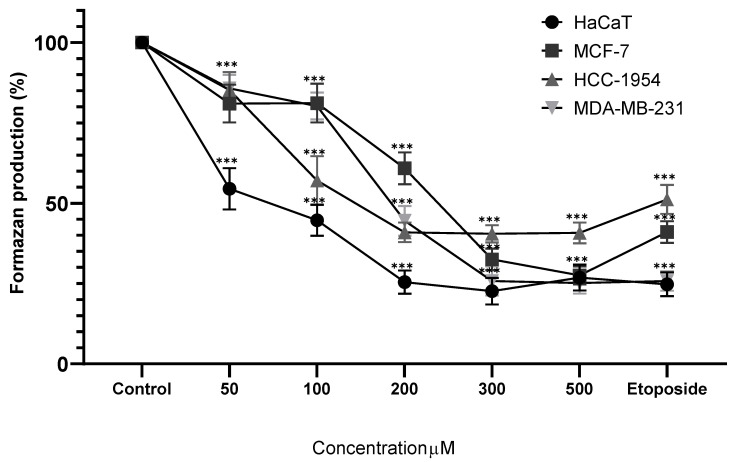
Percentage of formazan produced after treatment with Cv for 24 h. Etoposide (0.18 mg/mL = 305.8 µM) was used as cell death control. Determination was performed by MTT assay, and percentages were calculated in comparison to the untreated group, which was set as 100% formazan production. Results are expressed as mean ± standard deviation from three independent experiments performed in triplicate. Statistical analysis between groups was performed using ANOVA and Dunnett’s post hoc test (*** = *p* < 0.001, compared with control).

**Figure 8 nutrients-17-03089-f008:**
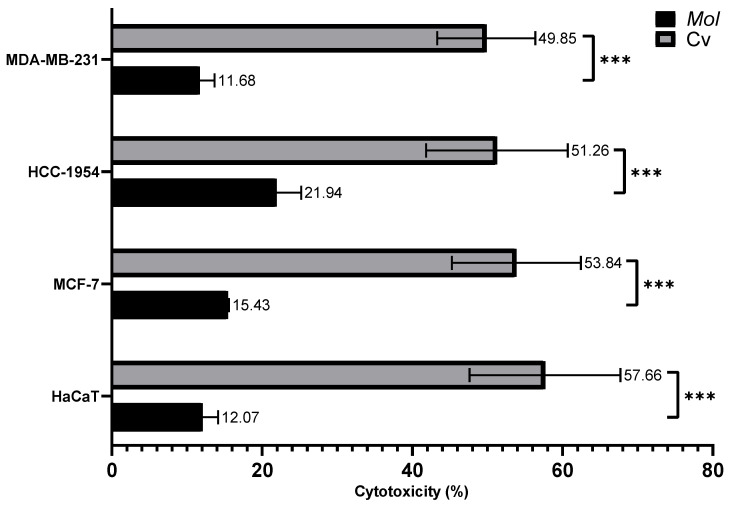
Cytotoxicity of MoI (48 h treatment) and Cv (24 h treatment) was assessed by quantifying LDH release in the supernatant of treated cells. Cells were exposed to the IC50 of both treatments. A control group treated with the lysis buffer included in the kit was used to define 100% cytotoxicity. Results are expressed as mean ± standard deviation of three independent experiments performed in triplicate. Mean comparisons between groups were conducted using Student’s *t*-test (*** = *p* < 0.001).

**Table 1 nutrients-17-03089-t001:** Bromatological analysis of the *L. graveolens*.

Nutritional Content	Per 100 g
Energy content	209 Kcal (874.4 KJ)
Moisture	7.8%
Ash	6.6%
Protein	6.8 g
Total fat	4.2 g
Available carbohydrates	36 g
Dietary fiber	38.6 g

**Table 2 nutrients-17-03089-t002:** Total phenolics and flavonoids content and antioxidant capacity (DPPH, ABTS, and FRAP) of MoI.

ABTS	DPPH	Total Phenolics	Total Flavonoids	FRAP
42.837 ± 0.175 mMEQ Trolox/g	66.303 ± 0.228 mMEQ Trolox/g	27.765 ± 1.095 mgGAE/g	22.343 ± 0.096 mg Eq quercetin/g	109.85 ± 0.51 mMEQ FeSO_4_/g

The results were normalized per gram of dry sample of *Lippia graveolens*.

**Table 3 nutrients-17-03089-t003:** IC50 of MoI and Cv in each cell line.

Cell Line	*MoI* IC50 (S.E.)	Cv IC50 (S.E.)
HaCaT	0.18 mg/mL (0.016)	110 µM (19.19)
MCF-7	0.08 mg/mL (0.006)	211 µM (18.95)
HCC-1954	0.15 mg/mL (0.010)	123 µM (6.09)
MDA-MB-231	0.17 mg/mL (0.024)	121 µM (10.85)

The standard error is shown in parentheses.

## Data Availability

Not applicable.

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
