# Peer review of "Evaluation of Antioxidant-Rich Mexican Oregano (Lippia graveolens) Infusion and Carvacrol: Impact on Metabolic Activity and Cytotoxicity in Breast Cancer Cell Lines"

_nutrients, 2025, doi:10.3390/nu17193089_

Round 1
Reviewer 1 Report
Comments and Suggestions for Authors
This manuscript presents an original and interesting study that compares the potential anticancer properties of Mexican oregano (Lippia graveolens) infusion and carvacrol in breast cancer cell lines. The authors examine the biochemical composition and antioxidant activity of the infusion, as well as its impact on metabolic activity and cytotoxicity in cancerous versus non-cancerous cells.
I have made some comments about how you could improve your work. This doesn't mean you have to agree or rewrite it the same way. It's just a suggestion and another way of looking at things to help you.
1) The 'Introduction' section contains strong global statistics on breast cancer, but it would be useful to briefly mention regional or national data (e.g. from Latin America or Mexico) to emphasise the relevance of studying Lippia graveolens specifically.
2) The 'Introduction' briefly mentions the WHO endorsement of complementary therapies. This could be reinforced by referencing recent reviews or meta-analyses on plant-derived compounds in oncology, which highlight the increasing demand for safe, affordable and accessible alternatives.
3) While the text acknowledges the limited evidence regarding aqueous oregano extracts, it would benefit from clearer differentiation between carvacrol (a single compound) and MoI (a complex phytochemical mixture). This would better justify the decision to compare Cv and MoI.
4) The first two paragraphs (the general definition of cancer and the description of breast tissue) are accurate, but they are slightly too long for a specialised audience. They could be condensed to focus more quickly on breast cancer subtypes and therapeutic challenges.
5) The final paragraph of the 'Introduction' section is informative but lengthy. Breaking it into two concise sentences (one addressing the knowledge gap and one addressing the study aims) would improve readability and emphasis.
6) Some subsections in the 'Materials and methods' section (e.g. MoI preparation, FTIR and HPLC) contain repetitive details regarding concentration, centrifugation and storage. These could be consolidated to avoid redundancy and improve readability.
7) Consider adding a schematic workflow figure summarising the experimental design (plant characterisation → antioxidant assays → cell assays). This would help readers to quickly grasp the workflow.
8) While the origin of L. graveolens is given, further botanical identification details, such as the voucher specimen number and herbarium deposition, would strengthen reproducibility and scientific rigour.
9) The methods for ABTS, DPPH, FRAP, total phenolics and flavonoids are clear. However, it would be useful to indicate the number of replicates performed for each assay and how the results were normalised (per g of dry weight or per mL of infusion).
10) While the HPLC method is detailed, it would be helpful to briefly indicate the limits of detection (LOD) and limits of quantification (LOQ) to reinforce reliability.
11) The MTT and LDH assays are well described, but the text could clarify why different incubation times were used for MoI (48 hours) and Cv (24 hours).
12) Information on how many independent experiments were performed and how many technical replicates were used per experiment should be included.
13) The section lists T-tests, ANOVA and Dunnett’s post hoc tests. It would improve clarity to specify which tests were applied to which datasets, and whether assumptions of normality and variance homogeneity were checked.
14) Units should be standardised throughout (μM vs µM, mg/mL, g/mL, etc.).
15) Some of the paragraphs in the 'Results' section are very long (e.g. 3.6 'Cell metabolism assay'). Breaking these up into smaller sections, one for each cell line or treatment, would improve readability.
16) Providing 95% confidence intervals for IC50 values would strengthen the rigour of the results.
17) While the 'Results' section focuses on raw data, it could benefit from brief statements summarising trends. For example, 'MoI exhibited lower cytotoxicity than Cv across all cell lines, with greater selectivity towards cancer cells'. This would bridge the gap between data and interpretation without moving into discussion.
18) Some of the paragraphs in the 'Discussion' section are very long (for example, those discussing FTIR, HPLC or Cv cytotoxicity). Clearly defining subtopics within smaller paragraphs would improve clarity.
19) While the antioxidant-cytotoxicity link is mentioned, more explicit discussion of the proposed molecular mechanisms, especially regarding the interaction of phenolic compounds or flavonoids with signalling pathways in BC cells, would be beneficial.
20) Providing a short summary table or figure comparing the IC50 values of MoI and Cv across different cell lines would help readers to understand the comparative results more easily. Similarly, a brief discussion of why MoI exhibits lower cytotoxicity than Cv could incorporate additional mechanistic hypotheses (e.g. synergistic interactions among multiple compounds or differences in bioavailability).
21) The 'Discussion' section does not explicitly mention any limitations, such as those relating to the in vitro study, the lack of in vivo data or variability due to plant origin, nor does it outline any future research directions. Including a paragraph on this would strengthen the section and provide context for further studies.
22) The current 'Conclusions' paragraph is long and dense, combining results, future directions and potential clinical relevance into a single sentence. Splitting it into two or three shorter sentences would improve clarity and impact.
23) While it mentions composition, antioxidant activity and cytotoxicity, it could briefly highlight the most important quantitative or comparative results (e.g. the effect of MoI relative to Cv or trends in IC50 across BC subtypes).
24) The 'Conclusions' section could also briefly acknowledge that this study is in vitro, and that translation to clinical settings requires caution. This would add transparency and scientific rigour.
25) While the mention of cytokines and redox enzymes is good, a more general statement on the need for in vivo studies or clinical validation could strengthen the conclusion. Optionally, a sentence emphasising the potential for combination therapies with conventional treatments could enhance its relevance.
26) Replace 'emerging as a promising field' with a more assertive phrase such as 'represents a promising avenue for the development of novel therapies'.
27) The 'Abstract' contains many sentences of over 50 words, which makes it difficult to read. Breaking them into two or three shorter sentences would improve comprehension. Consistency is key: currently, some phrases mix 'reduction in cell metabolism' with 'antiproliferative effect', which may confuse readers. Including a brief quantitative comparison (e.g. IC50 values or relative cytotoxicity) would also strengthen the abstract.
While this manuscript presents valuable findings, improving the clarity and depth of the discussion, as well as the methodological detail, would further enhance its impact. The findings are well supported by the experimental approaches employed. I recommend accepting the manuscript with some revisions.
Comments on the Quality of English LanguageThe English could be improved to more clearly express the research.
Author Response
Response to Reviewer 1 Comments
Thank you very much for talking the time to review this manuscript. Please find the details response below and the corresponding revisions, Comments and Suggestions for Authors,
Comments and suggestion for Authors
This manuscript presents an original and interesting study that compares the potential anticancer properties of Mexican oregano (Lippia graveolens) infusion and carvacrol in breast cancer cell lines. The authors examine the biochemical composition and antioxidant activity of the infusion, as well as its impact on metabolic activity and cytotoxicity in cancerous versus non-cancerous cells.
I have made some comments about how you could improve your work. This doesn't mean you have to agree or rewrite it the same way. It's just a suggestion and another way of looking at things to help you.
We appreciated such brilliant observations we made the suggested changes point by point.
- The 'Introduction' section contains strong global statistics on breast cancer, but it would be useful to briefly mention regional or national data (e.g. from Latin America or Mexico) to emphasize the relevance of studying Lippia graveolens
We are grateful for your suggestion and have included epidemiological information on national breast cancer date in Mexico. This are on lines 79-83 of the document resubmitted for a second review.
- The 'Introduction' briefly mentions the WHO endorsement of complementary therapies. This could be reinforced by referencing recent reviews or meta-analyses on plant-derived compounds in oncology, which highlight the increasing demand for safe, affordable and accessible alternatives.
Regarding this observation, we reinforce this comment with bibliographic information to meta-analyses, found in the line 94 of the document.
- While the text acknowledges the limited evidence regarding aqueous oregano extracts, it would benefit from clearer differentiation between carvacrol (a single compound) and MoI (a complex phytochemical mixture). This would better justify the decision to compare Cv and MoI.
Regarding this comment, we mention that, although carvacrol is largely a component of oregano, phenolic compound are important the antitumoral response and oregano contains these phytochemical compounds, among others. We included part of this information in the document on lines 113-120.
- The first two paragraphs (the general definition of cancer and the description of breast tissue) are accurate, but they are slightly too long for a specialised audience. They could be condensed to focus more quickly on breast cancer subtypes and therapeutic challenges.
We have made the changes according to your kind suggestion, as you can see in the lines 70-77.
- The final paragraph of the 'Introduction' section is informative but lengthy. Breaking it into two concise sentences (one addressing the knowledge gap and one addressing the study aims) would improve readability and emphasis.
We appreciate your kind comment. We have made the relevant changes to the last paragraph for clarity. These changes were made on lines 121 through 125 for clarity.
- Some subsections in the 'Materials and methods' section (e.g. MoI preparation, FTIR and HPLC) contain repetitive details regarding concentration, centrifugation and storage. These could be consolidated to avoid redundancy and improve readability.
We have made the necessary changes to the material and methods section to clarify this important section and avoid repeating multiple steps. All of these changes can be seen in paragraphs 166-171, in the paragraphs corresponding to lines 232-243.
- Consider adding a schematic workflow figure summarising the experimental design (plant characterisation → antioxidant assays → cell assays). This would help readers to quickly grasp the workflow.
Based on your interesting comments, we have included a flowchart as suggested, which describes the operational summary of the project. It can be seen in figures 323-326.
- While the origin of L. graveolens is given, further botanical identification details, such as the voucher specimen number and herbarium deposition, would strengthen reproducibility and scientific rigour.
We appreciate your comment and agree with your observation. In the supplementary material, we attach proof of identification of L. graveolens, obtained from Totaltiche Jalisco. The identification report is also included. Because the information is in Spanish we attach an English translation of the original document. This observation can be found on line 147-151.
- The methods for ABTS, DPPH, FRAP, total phenolics and flavonoids are clear. However, it would be useful to indicate the number of replicates performed for each assay and how the results were normalised (per g of dry weight or per mL of infusion).
Based on your kind comments, we would like to mention that all assays were independent and performed in triplicate for each analysis, as can be verified in the various paragraphs on lines 182-183, 189-190,200-201,208-210 and lines 216-218 of the document. Initially, all assays were performed in grams of dried oregano in dry weight, for the infusion to be prepared and subsequently used in microliters for the various analyses.
- While the HPLC method is detailed, it would be helpful to briefly indicate the limits of detection (LOD) and limits of quantification (LOQ) to reinforce reliability.
We appreciate your comment and inform you that the observation, have made in the document on lines 569-571, and 594-596, which explain in detail equation of the curves respectively.
- The MTT and LDH assays are well described, but the text could clarify why different incubation times were used for MoI (48 hours) and Cv (24 hours).
We appreciated your comment and would like to mention that un lines 386-405 we describe the reasons for our decision to use incubation times in the presence of Mol and Cv.
- Information on how many independent experiments were performed and how many technical replicates were used per experiment should be included.
Regarding this comment, we appreciate your observation and note that all experiments were performed in three independent trials and in triplicate, as described in the document on lines 284-285, likewise on lines 300-301 and finally on lines 313-314.
- The section lists T-tests, ANOVA and Dunnett’s post hoc tests. It would improve clarity to specify which tests were applied to which datasets, and whether assumptions of normality and variance homogeneity were checked.
Regarding the analyses performed for the variables, we are pleased to inform you that Student`s t-test was performed for the cytotoxicity assays. ANOVA, Dunnett`s and post-hoc test were performed to evaluated the time-response curves of cell exposed to Mol, as well as their metabolic response. The description is presented in lines 319-320 of the document.
- Units should be standardised throughout (μM vs µM, mg/mL, g/mL, etc.).
We appreciate your comment and inform you that we have carefully reviewed the document to establish this unit standardization. The modification has been included in 379, 441 and 463 lines.
- Some of the paragraphs in the 'Results' section are very long (e.g. 3.6 'Cell metabolism assay'). Breaking these up into smaller sections, one for each cell line or treatment, would improve readability.
Based on your suggestion, we made the changes to the results section, as can be seem in lines 405-437.
- Providing 95% confidence intervals for IC50 values would strengthen the rigour of the results.
We appreciate your observation and inform you that in lines 479-480, which corresponds to the identification of the IC50 of Mol and Cv, the standard error of the data shown in parentheses is presented in the table, verifying the reliability of the data.
- While the 'Results' section focuses on raw data, it could benefit from brief statements summarising trends. For example, 'MoI exhibited lower cytotoxicity than Cv across all cell lines, with greater selectivity towards cancer cells'. This would bridge the gap between data and interpretation without moving into discussion.
Based on our observations, the table attached shows the standard error on IC50 concentration for Mol and Cv concentration in the different cell lines 418-437 and lines 489-490. We appreciated your suggestions.
- Some of the paragraphs in the 'Discussion' section are very long (for example, those discussing FTIR, HPLC or Cv cytotoxicity). Clearly defining subtopics within smaller paragraphs would improve clarity.
Based on your kind suggestions, we made the necessary changes related to the discussion of FITR, HPLC, and cytotoxic. The relevant changes were made to the document, for your review see lines 544-559, and lines 575-599.
- While the antioxidant-cytotoxicity link is mentioned, more explicit discussion of the proposed molecular mechanisms, especially regarding the interaction of phenolic compounds or flavonoids with signalling pathways in BC cells, would be beneficial.
We appreciated your kind comment. This strong association is described more clearly in lines 600-624.
- Providing a short summary table or figure comparing the IC50 values of MoI and Cv across different cell lines would help readers to understand the comparative results more easily. Similarly, a brief discussion of why MoI exhibits lower cytotoxicity than Cv could incorporate additional mechanistic hypotheses (e.g. synergistic interactions among multiple compounds or differences in bioavailability).
Based on your observation, the concentration of Mol and Cv. With their respective standard error, have been included in the IC-50 table. This suggest why the isolated metabolite od carvacrol could more rapidly. However, since the Molt infusion contains a higher amount of phenolics, it profoundly modulated breast cancer cell, as explained in lines 661-663.
- The 'Discussion' section does not explicitly mention any limitations, such as those relating to the in vitro study, the lack of in vivo data or variability due to plant origin, nor does it outline any future research directions. Including a paragraph on this would strengthen the section and provide context for further studies.
Your kind comment makes un reflect on how importance it is to consider that this in vitro study would only allow us to observe the importance of herbal medicine, such as the use od lippie graveolens, which could be the starting point for analyzing experimental preclinical model and observing the probable metabolic effect of consuming this infusion, as can be seen in lines 704-706, and 709-726 of the document.
- The current 'Conclusions' paragraph is long and dense, combining results, future directions and potential clinical relevance into a single sentence. Splitting it into two or three shorter sentences would improve clarity and impact.
We are very grateful for your suggestion on writing the article`s conclusions. We have drafted the changes based on your suggestions, which can be seen on lines 736-743 of the research artic
- While it mentions composition, antioxidant activity and cytotoxicity, it could briefly highlight the most important quantitative or comparative results (e.g. the effect of MoI relative to Cv or trends in IC50 across BC subtypes).
We appreciate your comment, and what we observed is that the responses is independent in each cell lines in relation to the IC50 with Cv compated to non-tumorigenic lines, as described in lines 644-649 of the document.
- The 'Conclusions' section could also briefly acknowledge that this study is in vitro, and that translation to clinical settings requires caution. This would add transparency and scientific rigour.
We kindly made the suggested changes to this section and also inform you that in order to reinforce this comment in the final section of the discussion, corresponding to lines 727-729 and 736-743, we revisit these suggestions.
- While the mention of cytokines and redox enzymes is good, a more general statement on the need for in vivo studies or clinical validation could strengthen the conclusion. Optionally, a sentence emphasising the potential for combination therapies with conventional treatments could enhance its relevance.
We appreciate your kind comment. It is definitely important to mention that this research is in vitro, which only allows us to identify changes in the oxidative and cytokines component. It is extremely important to emphasize that it could be used as adjuvant therapy in breast cancer cells, and that is the beginning of research into in vivo studies to demonstrate the potential effect of Lippia graveolens. The changes made to the document can be found on lines 736-740.
- Replace 'emerging as a promising field' with a more assertive phrase such as 'represents a promising avenue for the development of novel therapies'.
Your kind comment has been enriching for our research work and has already been modified in line 741. We appreciate your observation
- The 'Abstract' contains many sentences of over 50 words, which makes it difficult to read. Breaking them into two or three shorter sentences would improve comprehension. Consistency is key: currently, some phrases mix 'reduction in cell metabolism' with 'antiproliferative effect', which may confuse readers. Including a brief quantitative comparison (e.g. IC50 values or relative cytotoxicity) would also strengthen the abstract.
We appreciate the reviewer for this valuable observation. The abstract has been thoroughly revised to improve clarity and readability by dividing long sentences into shorter ones. In additions we clarified the distinction between “reduction of cellular metabolism” and “antiproliferative effect” to enhance coherence. Finally a brief quantitative comparison including IC50 values has been incorporated to strengthen the summary. The changes can be seen in the lines 31-45
While this manuscript presents valuable findings, improving the clarity and depth of the discussion, as well as the methodological detail, would further enhance its impact. The findings are well supported by the experimental approaches employed. I recommend accepting the manuscript with some revisions.
We sincerely appreciate your positive evaluation of our research paper and for recommending acceptable with revisions. Following their suggestion, we have carefully revised the manuscript to improve the clarity and depth of the discussion as well as to provide additional methodological details that improve transparency and reproducibility.

Reviewer 2 Report
Comments and Suggestions for Authors
This work examines how an infusion made from Mexican oregano and a compound called carvacrol affects breast cancer cells, focusing on their metabolism and toxicity. The study evaluates the antioxidant properties of the oregano infusion and carvacrol. It investigates how these substances influence the metabolic activity of various breast cancer cell lines. Results highlight potential effects on cell viability and suggest possible therapeutic applications.
The study underscores the importance of assessing plant-based antioxidants, given their potential therapeutic benefits and minimal side effects. It measures both antioxidant capacity and cytotoxic effects on multiple breast cancer cell lines, providing a comprehensive understanding of their bioactivity. Identifying and quantifying key compounds like carvacrol enhances the scientific foundation for understanding the biological effects observed.
Figure 3 is illegible. It needs to be redone.
However, the research is primarily descriptive and lacks mechanistic experiments to clarify the molecular pathways involved. As an observational study, it cannot establish causality between antioxidant activity and effects on cancer cell metabolism. While correlations are noted, the study design does not support definitive conclusions about cause-and-effect relationships at the metabolic level in cancer cells. Additionally, results from cell lines may not directly apply to in vivo conditions or human physiology.
It would be beneficial to include a clear discussion of the study's limitations due to its observational nature. Explicitly acknowledging that the design restricts causal inferences and emphasizing that findings are mainly correlational would improve transparency and scientific integrity.
Specifically, the authors should:
- Clearly state that the study does not establish causality between the antioxidant-rich infusion or carvacrol and effects on cancer cell metabolism.
- Discuss how the observational and in vitro approach limits the ability to infer mechanistic or causal relationships directly.
- Recommend future research involving mechanistic, in vivo, or clinical studies to verify and expand upon these findings.
Adding this discussion would strengthen the manuscript's validity and help readers interpret the results within the context of its methodology.
Author Response
Response to Reviewer 2 comments.
Thank you very much for taking the time to review this manuscript. Please fin detail response below and the corresponding revisor.
Comments and Suggestions for Authors
This work examines how an infusion made from Mexican oregano and a compound called carvacrol affects breast cancer cells, focusing on their metabolism and toxicity. The study evaluates the antioxidant properties of the oregano infusion and carvacrol. It investigates how these substances influence the metabolic activity of various breast cancer cell lines. Results highlight potential effects on cell viability and suggest possible therapeutic applications.
The study underscores the importance of assessing plant-based antioxidants, given their potential therapeutic benefits and minimal side effects. It measures both antioxidant capacity and cytotoxic effects on multiple breast cancer cell lines, providing a comprehensive understanding of their bioactivity. Identifying and quantifying key compounds like carvacrol enhances the scientific foundation for understanding the biological effects observed.Figure 3 is illegible. It needs to be redone.
We are very grateful for your comments and observation regarding figure 3, which shows the quantification of some of the compounds in the infusion used for this research project. We have modified it for clarity and better visualization, as can be seen in line 363.
However, the research is primarily descriptive and lacks mechanistic experiments to clarify the molecular pathways involved. As an observational study, it cannot establish causality between antioxidant activity and effects on cancer cell metabolism. While correlations are noted, the study design does not support definitive conclusions about cause-and-effect relationships at the metabolic level in cancer cells. Additionally, results from cell lines may not directly apply to in vivo conditions or human physiology.
We appreciate your comment. In this project, we primarily evaluate the potential role of the infusion of Lippia graveoles (Mol) and carvacrol (Cv), in breast cancer cell and their metabolic and cytotoxic activity in an in vitro model. This would be a starting point for further evaluation of molecular mechanism and propose their study in preclinical models.
It would be beneficial to include a clear discussion of the study's limitations due to its observational nature. Explicitly acknowledging that the design restricts causal inferences and emphasizing that findings are mainly correlational would improve transparency and scientific integrity.
Specifically, the authors should:
- Clearly state that the study does not establish causality between the antioxidant-rich infusion or carvacrol and effects on cancer cell metabolism.
We appreciate your kind comment. This is in vitro study; does not allow us to establish causality between the antioxidant rich infusion or Cv and its effect on cancer cell metabolism. This statement can be identified on discussion section, lines 728-729 of the document.
- Discuss how the observational and in vitro approach limits the ability to infer mechanistic or causal relationships directly.
In accordance with your kind comment, we agree an in vitro test does not allow us to evaluate the metabolic and physiological mechanism of action of tumor cells in vivo, however, it allow us to know what the behavior would be with the presence of these natural compound such as Lippie graveolens and Carvacrol, based on these results in future research we will address the possible mechanisms at the molecular level, considering the antioxidant properties of the infusion and carvacrol. In addition, preclinical trials would be necessary.and in experimental animal before thinking that it an be applied to human. These concepts are discussed in lines 727-733.
- Recommend future research involving mechanistic, in vivo, or clinical studies to verify and expand upon these findings.
We appreciated your very positive comment and express that an our results lay the groundwork for future research using the natural infusion Lippia Graveoles and Carvacrol, which involves molecular aspect in breast cancer cell lines and stages in experimental model that allow us to evaluated metabolic aspects and observe the feasibility of their use, ass described in the line 727-733.
Adding this discussion would strengthen the manuscript's validity and help readers interpret the results within the context of its methodology.
We are very grateful for your comments and observation, which allowed us to enrich this research document.
